# A Leap from Negative to Positive Bond. A Step towards Project Sustainability

**Francesco Di Maddaloni [1],* and Roya Derakhshan [2,3]**

[1]   Project Management Research Centre, Kingston Business School, Kingston University, Surrey KT2 7LB, UK
[2]   Department of Management, Economics and Industrial Engineering, Politecnico di Milano,
     20156 Milan, Italy; roya.derakhshanalavijeh@polimi.it or roya.derakhshanalavijeh@alumnos.upm.es
[3]   Department of Engineering, Business Administration and Statistics, Universidad Politecnica de Madrid,
     28006 Madrid, Spain
*   Correspondence: F.dimaddaloni@kingston.ac.uk

**Abstract:** Departing from the normative formulation of stakeholder theory and building upon the current body of knowledge, this study aims to advance understandings on the benefits and challenges towards a more holistic approach to stakeholder management at the local level of major public infrastructure and construction projects (MPIC). Major construction projects, project managers, and the local community stakeholder are deemed to operate within a 'negative bond'. This interaction is elucidated through the attribution theory lens, which shape the conceptual framework of the study. This paper illustrates that a broader inclusiveness of stakeholder views into managerial decisions can initiate to break this negative bond and shift it towards positive relationships. An organization's conscious approach towards transparent communication with the local community stakeholder might help to shape a long-term perspective for better project benefits realization either at the national, regional, or local level. By 'exceeding stakeholders needs and expectations', project managers and local communities can better collaborate in order to achieve sustainable development over time.

**Keywords:** local community; attribution theory; communication; inclusiveness; sustainability

## 1. Introduction

Major public infrastructure and construction projects (MPIC) are important tools to foster modernization and enhance economic and social development (Altshuler and Luberoff 2003; Kara et al. 2016). However, MPIC projects have been notorious in terms of their level of aspirations, lead times, complexity, and divergent stakeholder interests. Despite the fact that more and larger infrastructure projects are continuously proposed and introduced, their growth results in an increased impact on people, budgets and urban spaces (Xue et al. 2015).

Although the likely benefits of MPIC projects are largely recognized, the uncertainty surrounding their impact and sustainability represents a key challenge for project managers. Typically, the decision making for these projects is not driven by the real needs of the society, but mainly by the technological, political, economic, and aesthetic sublimes highlighted by Flyvbjerg (2014) and, historically, MPIC projects have performed poorly in terms of benefits and public support due to their unavoidable impact on people and places (Bruzelius et al. 2002). It is by bridging the gap pertaining the poor performance of MPIC project, the lack of decision making, and poor public support, that this paper seeks to improve the sustainability of such projects through better inclusiveness of project stakeholders.

In fact, it is recognized that benefit realization and value co-creation are important elements for improving project performance (Laursen and Svejvig 2016; Ma et al. 2017; Vuorinen and Martinsuo 2018). Likewise, it is essential to minimize the waste of public resources by creating a better decision making

process that includes the needs and expectations of a broader range of project stakeholders, and that leads towards more sustainable construction projects. Notably, project managers are faced with increasing budget constraints, and thus the design, evaluation and selection of such highly costly projects has become particularly critical in turbulent economic conditions (Matti et al. 2017). Therefore, there is urgency for project managers to shape the most beneficial project for the real needs of the society, reducing public opposition and increasing benefit realization.

Projects as a vehicle for change play a crucial role in the sustainable development of organization and society, and recent debates have encouraged research in integrating broader societal objectives (sustainable developments) within projects (process and final goals) (Huemann and Silvius 2017). However, although it is widely recognized that minimizing benefit shortfalls and enhancing positive inputs is achievable through better stakeholder management procedures (e.g., Bourne and Walker 2005; Cleland and Ireland 2007; Olander 2007), academic thinking on MPIC projects seldom aligns project objectives with those of the local community (Choudhury 2014; Di Maddaloni and Davis 2018). Managers and academics alike have done little to achieve a human-centered vision for cities that enhances quality of life and produces prosperous neighborhoods that are sustainable over time. This has often created tensions and opposition at the local level of MPIC projects.

By looking specifically at the local community level and at the interconnections within which the project-based organization and local community interact, the literature discuss the negative impact that the local community can exert on project results (Aaltonen and Sivonen 2009; Olander and Landin 2005) and, consequently, MPIC projects (van den Ende and Marrewijk 2018; Xue et al. 2015). Unpopularity and local opposition are a common threat for MPIC projects whereby secondary and external groups try to influence the implementation of these projects (Boholm et al. 1998; Teo and Loosemore 2017).

Recent literature reviews in project and stakeholder management have elucidated that the majority of prior project research has focused on the management of those primary stakeholders important to the project's resources (e.g., Derakhshan et al. 2019a; Di Maddaloni and Davis 2017; Eskerod et al. 2015; Mok et al. 2015). Secondary stakeholders, which do not have a formal contractual relationship with the organization or direct legal authority over the organization (Eesley and Lenox 2006), seek a claim for a legitimate role in project decision making because they are risk bearers in such developments (Olander and Landin 2008) and, therefore, more time should be spent at the front end of a project (Pinto and Winch 2016) and developing a stakeholder engagement plan that includes a broader range of stakeholders (Derakhshan et al. 2019b; Eskerod et al. 2015; van den Ende and Marrewijk 2018). While stakeholder theory recognizes the growing importance of 'secondary actors' and, within them, communities as a new class of stakeholders (Crane and Ruebottom 2011), stakeholder management at the local level of MPIC projects is deemed to operate within negative dynamics. Recent research has shown the local community groups to be perceived as a risk by project managers and their engagement process often highlighted as a mere paid lips service, when decisions are already made and cannot be changed in any way (Derakhshan et al. 2019a; Di Maddaloni and Davis 2018; Teo and Loosemore 2017). This inevitably requires project managers to make extra managerial effort in terms of time and resources, such as building relationships through a 'names-and-faces' approach and transparent communication, also claimed by Hart and Sharma (2004).

An essential part of stakeholder management is stakeholder analysis, where project managers try to build the "correct" picture of stakeholder environment through an interpretation process (Aaltonen 2011). Organizations differ systematically in the modes and processes through which they interpret their external environment (Daft and Weick 1984; Milliken 1990). This interpretation process (perceiving, interpreting, and responding to changes) has been asserted to be at the heart of many stakeholder classification frameworks (Aaltonen 2011). However, although the literature has discussed the issues of community and project organization according to their impact exerting on each other, to the best of our knowledge, no research has discussed the way those two stakeholders perceive each other and the means that can be adopted to control their attribution from each other's behavior. In this respect, through the lens of the attribution theory (Jones and Davis 1965) the contribution of

this study is twofold. First, it helps to understand the benefits and challenges associated with the normative formulation of the stakeholder theory (e.g., Freeman et al. 2007), highly required in order to improve the performance of MPIC projects. Second, it sheds light on why these negative dynamics are manifested at the local level of MPIC projects, and how they can be shifted towards more positive relationships for better benefits realization. To that end, we formulated the following research question: *How can the local community stakeholders and project managers better control their attribution from each other's behavior in order to increase the sustainable development of MPIC projects?*

The outline of the paper is as follows. First, we investigate and review the current body of knowledge through literature studies of general stakeholder theory and on sustainability research. Second, we identify recognized challenges in current stakeholders and sustainability practices at local level of MPIC projects. Third, we use attribution theory to explain how the identified gaps can be tighten through more sustainable approaches. Lastly, we conclude by discussing implications for project stakeholder management, point future research endeavors, and highlighting the recognized limitations of the study.

## 2. Conceptual Background

### 2.1. Management of Stakeholders vs. Management for Stakeholders

Scholars have highlighted two different and contraposing stakeholder management approaches in current literature: the management of stakeholders and the management for stakeholders (Freeman et al. 2007). The first aligns to the instrumental formulation of stakeholder theory, which considers stakeholders as resource providers for the organization and categorizes them based on their potential ability to help or harm the organization (Eskerod and Huemann 2013). This approach is established on Salancik and Pfeffer (1978)'s work, which explains that stakeholders could be resource providers to the organization, based on their interests. Therefore, when there is a conflict of interest there must be trade-offs made, and whenever there is a need for trade-off, prioritization is unavoidable.

Instrumental theories are to provide tools and techniques to stakeholder management. The instrumental formulation of the stakeholder theory bridges between the management of stakeholders and achievement of corporation goals. In particular, the instrumental perspective requires the prioritization of organization goals over those of the stakeholders. In this formulation, stakeholders are considered as tools for achieving organization's goals, more than being interested legitimate groups or individuals. This links back to the three stakeholder relationship attributes introduced by Mitchell et al. (1997) of power, legitimacy, and urgency, which are not in line with the normative formulation of the stakeholder theory.

Freeman explains that: "*The stakeholder theory can be unpacked into a number of stakeholder theories, each of which has a 'normative core,' inextricably linked to the way that corporations should be governed and the way that managers should act*" (Freeman 2001, p. 133). Without the 'normative core' of these different standpoints, the stakeholder theory is misguided. The normative core of these stakeholder theories must answer to some questions to define the organization's responsibilities against stakeholders. Therefore, there is no one and only stakeholder theory, Freeman (2001) states, but a number of stakeholder theories each of which clarify one aspect of stakeholder management.

Among different narratives of stakeholder theory, feminist standpoint theory (Wicks et al. 1994) sheds light on the importance of caring by connections and relations with the stakeholders. Feminist thought is an effort to reduce the masculine influence of management studies on the stakeholder theory. It magnifies the importance of "*creating value for an entire network of stakeholders by working to develop effective forms of cooperation, decentralizing power and authority, and building consensus among stakeholders through communication to generate strategic directions*" (Wicks et al. 1994, p. 493). Therefore, value creation becomes a primary virtue, pushing competency among stakeholders as a second, yet important, honorability.

However, the often limited resources available within organizations have led to the predominance of the instrumental approach to stakeholder management in order to make stakeholders comply with the organization needs (e.g., Mitchell et al. 1997). In this perspective, the focus is narrowly on those vital stakeholders such as owners, suppliers, employees, and customers. In fact, literature is well documented on how managerial priorities have been given to those salient 'primary' groups or individuals that have a formal contractual relationship with the organization or direct legal authority over the organization (Eesley and Lenox 2006).

Management for stakeholders (Freeman et al. 2007), on the other hand, considers stakeholders as legitimate people or groups whose interests are respectful and valuable for consideration on their own. This holistic approach takes into account 'secondary' groups of stakeholders such as community groups, unions, consumer advocates, competitors, special interest groups, the media, and other non-governmental organizations (Aaltonen et al. 2008), regardless of their ability to help or harm the organization. As opposed to the instrumental approach, stakeholders are identified according to their interest in the corporation and not vice versa. In addition, the management for stakeholders approach explains that "*firms have a normative [moral] commitment to advance stakeholder interests and that this commitment shapes firm strategy and influences financial performance*" (Harrison and Freeman 1999, p. 480). The aim is focused on meeting and exceeding stakeholders' needs and expectations.

The normative formulation of stakeholder theory has an inclusive approach towards stakeholder management with a lucrative change in the perspective: organizational structures and general policies should be established according to the legitimate interests of all stakeholders (Donaldson and Preston 1995). That does not mean that shareholders would not be considered in value creation processes, but they must be considered as one of the stakeholders, beside others, whose interest is taken at the same level as other stakeholders (Phillips et al. 2003; Vuorinen and Martinsuo 2018). To summarize, the management for stakeholders view explains that corporations should care about the stakeholders and should consider their interests and concerns in their corporation activities. This opposes the management of stakeholders' view, which prioritizes organization interests by leading stakeholders to be aligned to their interests and needs.

These two readings from stakeholder theory will be used in this study. As mentioned above, we take the normative formulation as the authentic springboard for the discussion coming ahead. This will result in the elaboration of a conceptual framework illustrating the shift from the management of stakeholders towards the management for stakeholder approach. The shift is presented through the application of attribution theory and analyzing the perception that stakeholders make from each other.

*2.2. Why Stakeholder Theory Is Relevant to Project Management*

Cleland (1986) was the first scholar to identify stakeholder management as a core activity for enhancing better project performance. Since Cleland introduced the importance of stakeholder management into the context of temporary organizations, project management research and practice widely recognized that project stakeholder procedures are key factors in order to enhance positive input in projects (Bourne and Walker 2005; Cleland 1986; Cleland and Ireland 2007; Donaldson and Preston 1995; Olander 2007).

Stakeholder theory is a central part of the strategic management discourse (e.g., Buysse and Verbeke 2003; Minosa 2012). Nevertheless, the management of stakeholders in temporary organizations can be considered an established area in contemporary standards of project management (APM 2013; PMI 2013). Projects are social systems and their behavior through social interactions is highly influenced by the context in which they are embedded. By drawing on stakeholder theory, which is a recognized framework for analyzing the behavioral and cognitive aspect of the project management process, contemporary literature highlights stakeholder dissatisfaction as a root problem that causes many unsuccessful projects (Eskerod et al. 2015). According to Olander (2007) definition, throughout a project life-cycle a vast number of interests will be affected both positively and negatively; representatives of these interests are referred to as the project's stakeholders.

A recent study by Eskerod and Larsen (2018) elucidated how project management theory has drawn on reductionism, where a complex phenomenon is described in a simplistic way in order to grasp it. The reductionism approach is even more evident when exploring the stakeholder management field. In fact, despite many years of refinements, Crane and Ruebottom (2011) claim for a generic and artificial nature of stakeholder identification in traditional stakeholder theory which, according to McVea and McVea and Freeman (2005), requires moving away from the simplifications offered by 'role-based identification' and towards and a 'names and faces approach'. In this way, through identification as individuals with specific identities and interest, the moral value of stakeholders can be more easily recognized (Hart and Sharma 2004).

The over-simplified approach to the stakeholder conceptualization makes it difficult for project managers to predict stakeholder behavior and anticipate any detrimental attitudes. Mok et al. (2015) show how little details are provided by the various stakeholder analysis methods concerning stakeholder identification, classification, and assessment. Nevertheless, stakeholder literature suggests how stakeholder groups are generically identified and classified such as: fiduciary/non-fiduciary (Goodpaster 1991), primary/secondary (Clarkson 1995), proponents/opponents (Bonke and Winch 2002), core and fringe (Hart and Sharma 2004), actively/passively involved (Vos and Achterkamp 2006), direct/indirect (Lester 2007), external/internal (Aaltonen and Sivonen 2009).

Project managers have rarely looked at the project as embedded in the stakeholder's perception of the relevant past, present, and future in order to have a holistic approach within stakeholder analysis (Eskerod and Larsen 2018). In fact, much of the knowledge about stakeholder analysis practices in projects has been from the stakeholder impact perspective, especially on the impact that primary stakeholders can exert on project outcomes. This perspective, despite more than two decades of refinement and integration of stakeholder thinking into multiple disciplines, has led stakeholders to be defined mainly by their generic economic functions (Aaltonen and Kujala 2010), without the possibility to get richer and more profound insights on differing stakeholder perceptions, behaviors, needs, and expectations.

### 2.3. The Relevance of Sustainability to Stakeholder Theory

In the previous sections, we introduced two divergent aspects of stakeholder theory and its relevance to project management. In this section, we will explain how these core aspects of stakeholder theory are linked to sustainability. This is done through unpacking the concept of sustainability itself and building the bridge to connect this concept to stakeholder theory. Nevertheless, the relevance of sustainable development through the management for stakeholders approach is evaluated into the controversial and uncertain context of major infrastructure and construction projects, which depicts an interesting perspective for the study.

Sustainability is recognized as an emerging field of study (Huemann and Silvius 2017) where a conglomerate of definitions exists (Redclift 2005). It might not be possible to bring a final clarification for what sustainability is, but what can be done is to expand the concept in order to understand its origins and why its definition is linked to stakeholder theory. Although the roots of sustainable developments can be tracked back to the beginning of the eighteenth century (Morgenstern 2007), in contemporary times, wider attention has been given to the topic of sustainable development.

Regarded as a high-level objective in constitutional documents and official policies of states, regional, and local governments (Ji and Darnall 2018; Mossner 2016), sustainable development has been generically defined as "*development that meets the needs of the present without compromising the ability of future generations to meet their own needs*" (WCED 1987). In this definition, the values of solidarity and fairness between generations is thus evident.

However, recent literature emphasizes the need for a holistic approach that integrates ecological, economic, and social dimensions when making decisions in organizations and society (e.g., Aarseth et al. 2017). It was first in 1997 when Elkington introduced the triple bottom lines of sustainability as economic, social, and environmental. From Elkington's work, it is noticeable how

the ecological, economic, and social dimensions (planet, profit, and people) are interrelated and influence each other. In this respect, sustainable development aims at reconciling economic, social, and environmental efforts through the elaboration of more comprehensive long-term strategies and societies' wider involvement in decision making (Meadowcroft 2013; Rickards et al. 2014; Zeemering 2018).

The 1972 book, 'The Limits to Growth' (Meadows et al. 1972), concludes that the combination of global population growth and economic development would lead to depletion of natural resources. Dyllick and Hockerts (2002) emphasized that the balance between economic growth and social wellbeing has been around as political and managerial challenge for over 150 years. Therefore, especially in recent turbulent economic conditions, now more than ever, it is essential to minimize the waste of public resources by creating a better decision making process that catalyzes policy makers' resources and efforts (Greenspan 2004; Matti et al. 2017; NETLIPSE 2016).

Later, in 2013, Elkington incorporated the term of transparency into his definition of sustainability. The scholar explained that corporations are obliged to be transparent in their decision making and the way they impact (positively or negatively) their stakeholders, since nowadays it is not easy for them to hide their influences on their surrounding environment. Therefore, in line with Eskerod and Huemann (2013), the authors believe that sustainable development is a normative concept, where organization sustainability depends on the sustainability of its stakeholder relationships (Perrini and Tencati 2006).

Because stakeholder theory relies on various foundational moral approaches, its linkage with sustainability is thus evident. In order to achieve better performance, organizations have to strive for sustainable development through fairness and transparency in their decision making. Listening and considering stakeholders' needs and fears, embracing an open dialogue, and sustaining their expectations through an active involvement in the decision making process are prerequisites to shaping better organization long-term strategies and sustainable development. This is in line with the holistic approach to stakeholder management, which aims to exceed stakeholders' needs and expectations in order to achieve better performance and benefits realization over time.

## 2.4. Sustainability in Project Management

MPIC are drivers of change and, in this respect, their ability to 'sustain' the needs of the present without compromising those of future generations is somehow crucial. Novel debates are calling for sustainability as a new school of thought in project management (e.g., Huemann and Silvius 2017; Silvius 2017), where projects and their management are recognized as "a way to sustainability" (Marcelino-Sádaba et al. 2015).

By investigating the literature, Silvius and Schipper (2015) demonstrated that two relationships exist on the integration of sustainability and project management. The first, relates to the sustainability of the project's product such as the deliverable that the project realizes. Here, the focus is on operationalizing the 'three dimensions' suggested by Elkington (1997) of people–planet–profit by developing sets of indicators on the different perspectives with the scope of achieving a more sustainable deliverable (e.g., through specifications and design, benefits to be achieved, quality and success criteria). The second relationship relates to the sustainability of the project's process of delivering and managing the product. Here, the focus is on the integration of the dimensions of sustainability into the process of project management and delivery (e.g., stakeholder identification and engagement, procurement, risks, communication, and project's team selection).

It is recognized that in the process of developing and delivering the project, the two streams of the project's process and product interact (e.g., Labuschagne and Brent 2005). However, this study puts more emphasize on the relationship between sustainability and project management with respect to the sustainability of the delivery of the product (the process of realizing the product), than to the sustainability of the project deliverable (the product that the project realizes) (Kivilä et al. 2017). This emphasis is driven by the belief that through a sustainable process, project managers might better shape the 'right product' through interactions and engagement with those social actors operating in

the project environment. This suggests that project-based organizations must be able to continue their project deployment indefinitely.

Projects consume resources and influence their surrounding environments, so consideration of their sustainability cannot be done by isolating them from their surrounding ecosystem. Nevertheless, in recent years, project managers have faced legitimate pressure to demonstrate greater ethical responsibility in their decision making, requiring them to be attuned to the cultural, organizational, and social environments surrounding projects (Deutsch and Valente 2013). Therefore, it is expected to see projects in the future as an important social system for integrating sustainable development principles. Thus, it is important to integrate these principles into the business process of a company to receive performance benefits (Wagner 2007).

### 2.5. Stakeholder Management and Sustainability in Major Infrastructure and Construction Projects

Major public infrastructure and construction (MPIC) projects have offered a recurring theme for their controversy. Often cluttered by either 'delusional optimism' or misrepresentation, MPIC projects have been historically the target for political ambitions and public attention (Flyvbjerg 2014). The claim of enhancing the sustainability of large infrastructure development has therefore never been so actual like today.

Infrastructure spending is mainly driven by large-scale projects (Flyvbjerg 2014) and, although the global infrastructure market is predicted to continue growing between 6% and 7% yearly until 2025 (PwC 2014), there has been a negative focus in academic literature of large infrastructure developments, as they are too often unable to meet basic targets of budget, time, and expected benefits. According to Jia et al. (2011), MPIC projects have close connections with globalization and civilization. Nevertheless, these projects are the means of urbanizing the globalization (Moulaert et al. 2003). However, the poor performance recorded in these large developments, explains the inability of managers and project promotors with invested interests to manage globalization and city growth (Elmlund 2015). This inability is due, in part, to a lack of accountability, transparency, and wider involvement of society in the project decision making process.

Decision made by project managers have a significant impact on the strategic value delivered by major programme in the construction industry (Eweje et al. 2012; Vuorinen and Martinsuo 2018). However, organizational strategy frequently fails to achieve the desired results and, historically, these projects have performed poorly in terms of benefits and public support due to their unavoidable impact on people and places (Bruzelius et al. 2002). Therefore, for projects to meet and exceed stakeholders' needs and expectations, project management must be done in the context of sustainable development.

By recognizing the positive role that stakeholder management plays on project performance, this study adheres to the four important principles presented by Gareis et al. (2013) in order to apply sustainable development perspective in managing project stakeholders. (1) Considering values of decisions such transparency, fairness, and participation; (2) considering and balancing the project stakeholders' economic, ecologic, and social interests; (3) designing appropriate strategies to consider short-, medium-, and long-term stakeholders' perspectives; (4) delivering value to stakeholders at the local, regional, national, as well as global level.

Based on the above discussion, it is immediately noticeable how MPIC projects have historically struggled to adhere to those important principles highlighted by Gareis et al. (2013). Although the management for stakeholders approach fits betters those underpinning assumptions, little has been done by practitioners and academics alike to establish a holistic approach to stakeholder management, in order to deliver the promised benefits of MPIC projects either at the local, regional, or national level.

The focus on MPIC project benefits has been from the national government's or the large public or private organization's perspective (Mok et al. 2015). Therefore, Di Maddaloni and Davis (2017) called for an in-depth investigation at the local context of these projects and related stakeholder management practices addressed to local communities. Although the local community inputs remain not well perceived by project managers (Aaltonen and Kujala 2010; Derakhshan et al. 2019b; Di Maddaloni

and Davis 2018), major steps have been made in recent years through consultative processes such as the social impact assessment (SIA) or the statutory planning act, which, in the UK, is a compulsory pre-requisite for project approval. Nonetheless, NETLIPSE study has shown promising examples of how organizations have seen local stakeholder's involvement as valuable and beneficial in any project (Van Buuren et al. 2012; Hertogh and Westerveld 2009; Hertogh et al. 2008; McVea and Freeman 2005). Good examples are the Lisbon-Porto High Speed Line, the West Coast Main Line in UK, the Øresund Crossing in Denmark, the North/South Metro line in the Netherlands, and the Bratislava Ring Road.

However, local opposition is not uncommon in MPIC projects (Boholm et al. 1998). Due to the perceived benefit shortfalls of MPIC projects, well-organized actions from 'secondary stakeholders' groups have led to delays, cost overruns, and significant damage to the organization's reputation (e.g., Hooper 2012; Letsch 2013; Teo and Loosemore 2017; Watts 2014). Therefore, it is recognized that a broader inclusiveness of secondary stakeholders, such as the local communities who could be harmed by the organization's strategy, is required to enhance the performance and sustainability of MPIC projects.

Illustrating stakeholder management practices and correlated benefits at the local level of these projects will offer practitioners the opportunity to increase public support and reduce local opposition. These practices will be evaluated in the next section, utilizing the attribution theory lens.

## 3. Theoretical Lens: The Attribution Theory

This section aims at explaining the perception of the stakeholder local community and project managers from each other, while analyzing the humans' cognition process towards which perceptions are shaped.

Frequently used by social psychologists, despite the tremendous potential to explain a wide range of workplace behaviors, attribution processes have been underutilized in the organizational sciences (Martinko et al. 2011). This is quite surprising as attributions are reliable predictors of human behavior (e.g., Martinko et al. 2007) and essential to effective stakeholder management. Therefore, this study introduces attribution theory as its theoretical lens in order to explain how individuals attempt to understand the behaviors of others by attributing feelings, beliefs, and intentions to them.

By considering the perceptions that project managers and local community make from each other, we observe that on the one hand the stakeholder local communities, just like any other humans, work as naïve psychologists who observe organization's behavior, analyze it, and make their judgment about it. On the other hand, the same cognition process is happening in project managers' brains and these two processes make a bidirectional perception, which must be analyzed in order to understand how the relationship between these two groups of stakeholders can be improved by overcoming current barriers.

What emerges from theory and practice is that most of the classifications in the literature see the local community (and arguably any other group) represented as a single entity. This over-simplified classification is often described with a broad-brush approach which, according to Dunham et al. (2006, p. 24) "*ignore or fail to take account of important and marginalized interests*" of many autonomous units of people with their own needs, fears and expectations (Teo and Loosemore 2014). The local community, for instance, is often managed as a unified entity with quasi-similar interests, concerns, harms, and helps. By looking at local community as a group of individuals with different ages, genders, cultures, demands, and interests, we reach some new individually specific attributes that must be considered while engaging with stakeholders.

Accordingly, in order to be able to have a better understanding of stakeholders as real people, we must analyze the perceptions that the stakeholders make from each other. This aim leads us to apply attribution theory as the theoretical lens.

Jones and Davis (1965); Weiner (1972); Kelley (1973) and other psychology researchers explained that attribution is a rational process in which humans draw conclusions about others' behavior. As Taylor and Fiske (1978) explained, individuals become satisfied with the first satisfactory cause

that they find. Therefore, in the case of availability of different explanations for a behavior, the first salient one should be selected as the cause. These salient sources could be either distinctive qualities of the agent (e.g., the quality of the project manager accountable for the project-based organization) or a particular cause existing in observer's memory. Feldman (1981) suggests that previous categorization of the trait or personality of the organization performing an act is considered as a very salient source of causes for observers.

Feldman (1981) also suggests that assigning characteristics to agents can be controlled. A controlled categorization process is triggered when incoming information reaches a threshold of the discrepancy. The meaning of Feldman's conceptualization is that when observed behavior makes a sufficient distance from the observer's assumed prototype or expectations, more information should be gathered and a new decision made (Feldman 1972). Similarly, Lord and Smith (1983) explained that the level or amount of information processing is a significant factor in the attribution process. The rule of thumb here is that when the level of available information is high, a conscious and controlled attribution is made. This is very much connected to the concept of 'shadow of the context' claimed by Eskerod and Larsen (2018), and the need of bringing more information into the project conversation in order to better understand the stakeholder's perceptions.

In the lower levels of available information, the decision is made automatically without high consciousness. As explained by Derakhshan et al. (2019a), during the conscious mode of decision making of individuals, the judgment about the organization is based on the information received from the organization as well as government, media, and other interested social groups. However, when this decision is made by the community, this primary judgment would be used to guide interpretation of any new information received. The new information is considered as consistent with the existing judgment and as new evidence to confirm the former decision.

This study uses attribution theory as a tool to extend the current body of knowledge surrounding the normative formulation of stakeholder theory. This will inform the management for stakeholders approach, which we consider highly necessary to enhance sustainable development over time. By understanding how perceptions are made and how these perceptions can influence the stakeholders' behavior in communicating with each other, current stakeholder management procedures are elucidated through attribution theory as a lens that shed light on the way the local community stakeholder and project managers interact and perceive each other in MPIC projects.

## 4. Analysis and Discussion

### 4.1. A Negative Bond

Research and practice have offered evidences for the existence of a negative bond between local community and project organization, more specifically project managers. This bond is shaped from the bidirectional perceptions that these two groups of stakeholders make from each other's behavior.

The existence of local community in the close proximity of a project could be a source of problem for the project managers. There are not many successful examples of organizations being able to tackle the issues with communities through constructive communication. There is vast literature that recognizes the (negative) impact that the local communities stakeholder can exert on project outcomes and how MPIC developments are often seen as a threat rather than an opportunity (e.g., Aaltonen and Kujala 2010; Boholm et al. 1998; Bornstein 2010; Newcombe 2003; Olander and Landin 2005, 2008; Teo and Loosemore 2014, 2017). Accordingly, the negative dynamics, in which secondary stakeholder management operates at the local level of MPIC projects, is also reflected in the project management perception of the local communities' stakeholder.

Focusing on the construction industry, a recent empirical study by Di Maddaloni and Davis (2018) highlighted the benefits associated with an inclusive stakeholder management approach. By interviewing experienced project managers directly involved in the management of 'secondary stakeholders', the authors tried to understand the perceptions that construction project managers have

in regard to the stakeholder local community in MPIC projects. These perceptions were captured and contextualized by reflecting on 15 MPIC projects in the UK.

Based on the perceptions of the interviewees, Di Maddaloni and Davis (2018) indicate that project managers of megaprojects often perceive the construction impact at the local level negatively. This is mainly associated with the disruption that these projects typically have in individuals' day-to-day lives (e.g., pollution, traffic congestion, land acquisition, changes in landscape).

Organizations often do not want their project managers to deal with the external world and they are primarily looking at the local communities as a risk from a perspective of the implications of delays around public consultation (Di Maddaloni and Davis 2018). Therefore, it is important to discuss how project managers coming with this mentality of seeing local community as a threat to the project and a burden to its achievements result them in considering a predetermined personality or 'trait' for the local community. They will, therefore, categorize the local community in one unified entity with some salient characteristics and roles. This shifts the project managers' cognition process from conscious to automatic (Feldman 1981), and would tremendously narrow the efficiency of the communications with them. Any piece of information transferred during communication with local communities would automatically be processed to be just another evidence to proof the community's trait: a group that is an obstacle to the project. This, as we argue, limits the exploration of opportunities of co-creating values with various individuals by exploring their characteristics (McVea and Freeman 2005).

The exact similar issue exists the other way around where the community's observation, analysis, and perception are colored by the personality they consider for the project organizations in general (Derakhshan et al. 2019a). Often the stakeholder local community believes that the implementation of MPIC projects in their proximity would result in environmental degradation, financial issues, and disruption in their day-to-day lives. This will isolate them and perpetuate the perception that the project organization will not allow them to make decisions that will influence them in the first place.

On the other hand, communities negatively affected by construction projects are becoming increasingly empowered, organized, and willing to engage in protest (Teo and Loosemore 2017). Inevitably, this will reinforce the negative perception that project managers have from the community. Organizations often do not adhere to the concept of 'shadow of the context' to better understand the stakeholder's perceptions of the relevant past, present, and future by bringing more information into the communication strategy deployed by the project manager (Eskerod and Larsen 2018). By not providing high level of information to their stakeholders, they do not allow them to effectively control their conscious attributions and perceptions, resulting in organizations' unsustainable approach towards communities and excluding them from the decision making process. This represents the point when a negative bond of perceptions is shaped between the project organization and the local community.

The above discussion brings many challenges in any shift towards a community-inclusive approach that requires the identified negative dynamics to be broken and changed into positive ones, in order to achieve better benefit realization and sustainable development over time. Figure 1 shows the negative dynamics that occur in MPIC projects between the local communities and project managers.

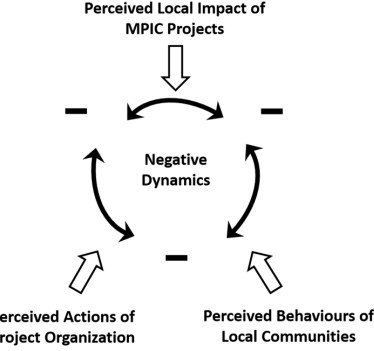

**Figure 1.** Negative dynamics at the local level of MPIC projects, adapted from Di Maddaloni and Davis (2018).

### 4.2. A Shift

The abovementioned negative bond is the result of the presumed personality or trait that the two perceivers (i.e., community and project managers) assign to each other. In order to break this negative bond, there is a need to change their perception; however, this is not an easy task. This process calls for shifting the community's and organization's cognition process to a conscious one, where both sides leave behind their negative perceptions in order to collaborate in co-creating values through project deployment.

Referring to the principles of the attribution theory (Kelley 1973), the aim is to answer the question: "how would this shift from automatic to conscious attribution process happen when humans are naturally more willing to use the most accessible piece of information in their mind for assigning characteristics to others?". Feldman (1981) explains that a controlled categorization process is triggered when incoming information reaches a threshold of the discrepancy. Tost (2011) calls this a jolt and it happens when individuals' observed behavior is sufficiently distant from their assumed prototype or expectations, and more information should be gathered and a new decision made (Feldman 1972). Here, we suggest that there are two factors that organizations can adopt in order to start breaking this negative bond. The first factor for triggering a conscious observation is the inconsistency between the new actions and the individuals' expectations (Lord and Smith 1983), while the second factor is to overcome the lack of transparency in communication and involvement with the community.

More conscious evaluations would be made when relevant information is easily available and in the appropriate forms for the active observations (Lowe and Kassin 1980). This represents a call for the organization to work towards a more sustainable approach. Assuming that community evaluates the organizational behavior with the criteria taken from its societal values and norms (Tost 2011), and remembering that according to Henriques (2013) sustainable organizations have more societal compatible behaviors, we can conclude that organizations that strive for sustainable development would be judged more positively. This shift in the behavior, however, must be observable enough to provoke community to reconsider their former judgment of the organization (Derakhshan et al. 2019a). This links the debate to the ecological reading of stakeholder theory (Freeman 2001), which is about caring for the environment and merging business with ethics in both short-term and long-term perspectives.

This initial step must be accompanied by transparent communication and involvement with the community. Transparency and availability of data would result in both local community and project managers developing attributions that are consciously built over their present observations from the project practices. This will result in elimination of any rumors about organizational activities since all information are clearly transferred between actors. This supports the discussion, stemmed from moral philosophical stances of stakeholder theory, which organizations should willingly be transparent with stakeholders since it is the stakeholders' right to know about the process it takes for the decisions to be made, as well as the influences of those decisions on their life. Nevertheless, additional emphasis is put on the importance of communication and building relationships with the stakeholders, stemmed from the feminist reading of stakeholder theory (Wicks et al. 1994).

These initial steps taken by project managers would result in community abandoning their former judgment and processing their new observations for making a new perception towards a better vision of MPIC projects and the way the promised benefits are delivered and materialized. Local communities will be more willing to communicate and collaborate with project managers and their newly adopted behavior, because encouraged by a transparent and inclusive approach of the organization. This new approach would result in breaking the project managers' negative perception from the community, and increase the sustainability development of MPIC project at the local community level.

### 4.3. Achieving Sustainability through the Management for Stakeholders Perspective

The application of attribution theory has supported our debate to shift from the management of stakeholders to the management for stakeholders approach in order to prepare an effective inclusive and collaborative atmosphere in which 'secondary stakeholders' and project managers are making

positive perceptions from each other. Our discussion illustrates that the collaborative value co-creation, trustful relationship mentioned by all of the advocates of the stakeholder theory is not a myth. It can be a reality achieved by practitioners adhering to the normative formulation of stakeholder theory, through conscious consideration of sustainability, building relationship and trust with the wider community, and embracing their values as project targets (e.g., Ma et al. 2017; Eskerod and Huemann 2013; Freeman et al. 2007; Freeman and McVea 2001).

While these factors seem almost clear and straightforward, they have a hidden contradiction. Organizations that have a sustainable approach are more willing to be transparent towards their stakeholders, including the local community, while organizations with lower consideration of sustainability are naturally less willing to be open to the stakeholders. In addition, practitioners need to question if an organization claiming to be transparent with their local surrounding environment is sharing both its negative and positive influences, or it is just promoting the benefits it has created for the community.

While organizations must minimize their negative influences on the surrounding environment and move towards sustainability, they should also be transparent with stakeholders about their impact (either positive or negative). Being transparent would push the local community's cognition process to a conscious mode, from one in which they analyze only what they observe, and it reduces the unwelcomed rumors about the organization, which may not necessarily be correct. Instead, the project organization would provide more and correct information to them in order to better understand their perceptions and needs (Aaltonen 2011; Eskerod and Larsen 2018).

Managing the needs and expectations of project stakeholders falls under the responsibility of the project manager. It is recognized that balancing the stakeholder's economic, ecologic, and social interests through fairness, participation, and value creation might bring considerable benefits in order to achieve better project performance (Eskerod et al. 2015; Vuorinen and Martinsuo 2018). However, the holistic approach towards a broader inclusiveness of project stakeholder should not weaken the focus on 'vital' stakeholders. A wise balance to include the views of both the 'primary' stakeholders essential to organization survival and those legitimate 'secondary' stakeholders is a challenging yet crucial task for project managers, in order to achieve sustainable development and be attuned to the often limited resources available within project-based organizations. Nevertheless, the level of information and communication provided to project stakeholders and the local community should not escalate expectations that the organization cannot adhere to over the course of the project life cycle. Practitioners have to be attuned with a transparent communication flow that does not induce stakeholders' disappointment due to impossibility of embracing the often-conflicting wishes and expectations.

## 5. The Conceptual Framework: From Negative to Positive Bond—The Shift Presented

We have used attribution theory as a tool to extend the current body of knowledge surrounding the normative formulation of stakeholder theory. This theory has enlarged our understanding on the way the stakeholder local community and the project managers perceive each other's behavior, and how these perceptions result in having an influence on current stakeholder management practices and long-term sustainability of MPIC projects.

This study tried to elucidate why the negative bond between stakeholders exists, and how it can be changed into a positive relationship. This has informed the management for stakeholders approach, which we consider highly necessary to enhance sustainable development over time. Pointing to the commitment of the project-based organization in meeting or exceeding their stakeholders' needs and expectation, the focus was given to materialize the promised benefits at the local level of MPIC projects. The aim was to shed light on the lack of knowledge pertaining the local context in which these projects operate. The attribution theory explains why, to date, the broader inclusiveness of the local community into managerial decision making remains marginal and why this inclusive approach must be enhanced through transparent communication and close relationships with local communities.

The conceptual framework for the study is presented in Figure 2. Under the lens of the attribution theory, the conceptual representation of the study shows the shift from negative to positive bond, required in order to achieve better MPIC project sustainable development over time.

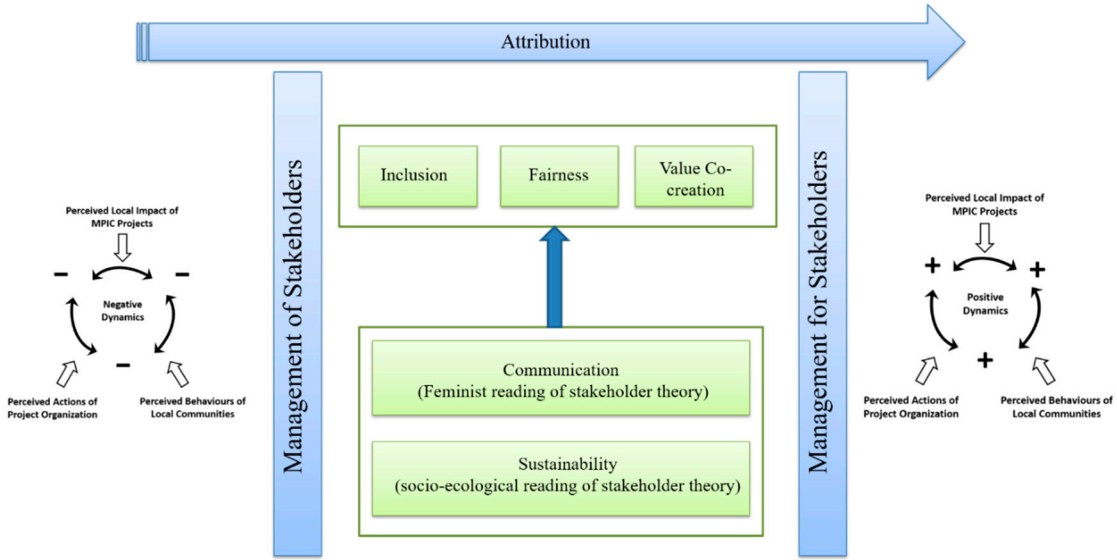

**Figure 2.** The shift from negative to positive bond.

These two straightforward, yet complex, steps of transparency and inclusiveness will eventually engage both community and project organization in value creation and crating the desired positive bonds. The important point is that the shift towards positive bond must be initiated by the project organization. In the light of achieving sustainable development through an inclusive approach, transparent communication, building trust, and sharing fair amounts of value with local communities, the two stakeholders would make true perceptions from each other. "*Of course, the firm does not have perfect knowledge, but when it discovers some danger or runs afoul of new competition, it is expected to inform the local community and to work with the community to overcome any problem*" (Freeman 2001, p. 131).

Freeman (2001) explains that people assume something is fairer when they have enough information about it. Thus, this communication would make local community believe that the condition is fair and the organization is trustful and legitimate. As Parsons (1968) explains, a legitimate organization is accepted by the larger society to exist and to continue to import, transform, and export energy, material, and resources from the surrounding environment. Thus, this communication can bring benefit to both groups of local community and project organization.

In this way, project managers can better transfer and communicate the positives of MPIC projects to the wider community, creating the right vision for such projects and improving performance; therefore delivering benefits either at national, regional, or local level by challenging people's negative perceptions. This requires a supportive organization culture that should be sustained by top senior/strategic management level to be effective.

Speaking of communications at the local level, we must remember that the notion is supported by the cognition processes of the humans as well as the feminist re-reading of stakeholder theory, which emphasizes the importance of a names-and-faces approach to stakeholder management. Therefore, this is embedded in the suggestion of transparent communications for reaching the desired positive bonds at the local level of MPIC projects.

## 6. Conclusions

By highlighting the importance of project stakeholder management to achieve sustainable development over time, this study aimed to understand how the local community stakeholders

and project managers can better control their attribution from each other's behavior in order to increase the sustainable development of MPIC projects. The answer was provided by digging into two different stakeholders' perspectives: the project organization and the stakeholder local community. These perspectives were investigated in the context of MPIC projects in order to elucidate how these two stakeholders make perceptions from each other and overcome the barriers that have historically seen MPIC as a 'built-in recipe' for local impact but not local benefits.

The results of our conceptualization revealed that managing the connections and relationships with the local community is more direct under the conditions of transparency in which the organization tries to minimize the unwanted influences of the project on the surrounding environment. This study sheds light on the importance of relationships and connections with the stakeholders and therefore is a contribution to the normative standpoint of stakeholder theory. The study suggests that transparency and inclusiveness at the local level are the main tools for improving the relationships with community through influencing their perceptions. This study concludes that sustainable development requires the organizations to be transparent and inclusive to their community also in conditions where the project impacts are not positive.

Noted limitations are that this paper is conceptual in nature and requires further empirical work. At present, it solely draws on inputs identified in literature on stakeholders, sustainability, and project management, as well as on logical deductions. The literature does, however, support our claims on the limitation of the instrumental approach to stakeholder theory and the need for a shift towards a holistic approach to stakeholder management (managing-for-stakeholders) that is required to overcome, at least partially, the lack of public support and benefit realization of MPIC projects at the local community level. Another recognized limitation is that the study discusses the negative dynamics at the local level of MPIC projects per se, without taking into considerations the interactions between these processes in different phases of the project life cycle.

Therefore, it is suggested that future efforts of scholars might build upon this study so that they can complement the presented research and expand current knowledge of how project-based organizations might enhance the inclusiveness of the local community and thus the long-term benefits and sustainability of MPIC projects. We do not consider our ontological and epistemological standpoint as limitations to the paper, but their underpinning assumptions can be challenged by authors with different philosophical perspectives. More specifically, we invite advocates of the instrumental formulation of stakeholder theory to reveal their discussion in the same context. The discourse may flourish the stakeholder theory.

**Author Contributions:** F.D.M. individual contribution can be summarized as follow: literature review and conceptualization of existing body of knowledge; analysis and conceptual framework; writing—original draft preparation writing-review and editing. R.D. individual contribution can be summarized as follow: literature review and conceptualization of existing body of knowledge; analysis and conceptual framework; writing—original draft preparation, writing-review and editing.

**Funding:** This research received no external funding.

**Acknowledgments:** The authors wish to acknowledge the anonymous reviewers and the editor for their valuable guidance.

**Conflicts of Interest:** The authors declare no conflict of interest.

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
