# Peer review of "A Leap from Negative to Positive Bond. A Step towards Project Sustainability"

_admsci, doi:10.3390/admsci9020041_

Round 1

Reviewer 1 Report

Thank you for offering me the opportunity to review this paper, which I found highly interesting – this piece of work offered me novel ideas and insights that are useful also in my own work. The reflections on the role and linkages between project sustainability management and stakeholder thinking were valuable, as well as the ideas on utilizing attribution theory to understand the sensemaking and interpretation processes of project managers and stakeholders. What I also valued was the fact that you aim to analyze both the perspectives of PMs and stakeholders and through this understand the complex interactional processes of these “two” groups (which are, however, many times characterized by high degrees of multiplicity).  

In the following, I discuss some areas and ideas for improvement.

First of all, I think you could be more specific with regard to the concept of secondary stakeholders in your introduction. The definition of secondary stakeholders is not quite straightforward in prior literature, and you could somehow explicate what your understanding in this paper is on this concept. I found the discussion on the importance on the authenticity of stakeholder “management” highly relevant in your introduction. I was, however, looking for more references to recent work on stakeholder engagement, which, at least to my understanding, better captures the management for stakeholder ideals as well as the value co-creation perspectives.  That is to say, I think that although the message of the failures of complex projects/large infrastructure projects in their stakeholder management is to certain extent true, I think that the real life situation might not be that black and white: we have a number of very successful projects, that have adopted a value co-creation approach with regard to secondary stakeholders and have delivered excellent results e.g. in terms of the innovation outcomes. The culture is changing (slowly though), and even though I can see the importance of highlighting “the failure focus” from the perspective of justifying the importance and uniqueness of your study, I think you could also give more recognition to the work that has focused on the engagement perspective and the positive dynamics and outcomes within the area of project stakeholder management. Here, potentially introducing some relevant procedures (e.g. social impact assessment, which as a consultative process can bring very valuable outcomes, and encourages in early dialogues with stakeholders) might be of value.

There is also some work at least on the way project managers perceive, interpret and enact their stakeholder environment, which has claimed that also PMs differ in their interpretation and approaches toward external stakeholders (Aaltonen, 2011). This relates closely to your attribution approach and may be useful for you.

You are also stating that “no research has discussed…”. In today’s world, full of researches and studies, we can rarely be certain that “no research has discussed”. Maybe you could be more modest here and milden your statement a bit e.g. that “to the best of our knowledge, no research..”.

It would also be really great, if you could clearly explicate your research question in the Introduction section).

The literature review is well written and delivers many valuable ideas and thoughts. Taking the holistic theme of the study and the dual focus on stakeholder management and sustainability, it is understandable that it is a bit fragmented. It could also be, that you could focus more clearly on the theme of stakeholder engagement, cognitive processes and attribution theory in particular. In this scenario, the review of e.g. stakeholder theory could play a significantly minor role. However, on the other hand, I enjoyed reading your “sensemaking” on stakeholder literature, which was fresh and provided interesting insights.

Coming back to my earlier comment, maybe you could also provide positive perspective and attempts in terms of stakeholder engagement in your literature review (e.g. in many projects SIAs are a highly useful and value producing arena for early stakeholder engagement and consensus building). At the moment, the picture that is portrayed on the state-of-the-art/practice of secondary stakeholder management is relatively negative. However, I do agree with you that prior research has mostly focused on the negative side and failure cases.

While I learned from your literature review, that there is complexity within/inside the complex groups of stakeholders (referring to the multiplicity of stakeholders concept) and that more focus should be placed on individuals, identities, interests (and maybe also potentially on issues that drive stakeholders), I was not able to identify this thinking so strongly in your final model (Figure 3). Your model, at least to me, also seemed to reinforce the attribution of the groups in to their role-based/function-based roles (project managers and local communities).  This is maybe still something that you could re-consider.

I very much liked your ideas on applying the attribution theory in order to facilitate our understanding of the interpretation processes of project personnel and local communities/secondary stakeholders. This perspective reinforces and follows the research stream that has called for more understanding on the actual cognitive processes of decision-makers and stakeholders alike. I would, however, still urge you to reconsider the conceptual framework. I can see the point in this positive/negative categorization, but what about the different shades of grey, as well as interactions between these processes during the project lifecycle? I think that the relationships and interpretations can also be multi-faceted and even issue specific within projects and both negative and positive collaborative dynamics (and interpretations) may co-exist at the same time. What kind of implications would this kind of approach have for your model? Maybe this is something that you could also consider in the areas for future research.  Your Figure 3 also seems to have some kind of system dynamics thinking behind it (the re-enforcing cycles). Is this a correct interpretation and should it be made more visible and explicit?

Finally, there are some typos in terms of the references e.g. it should be Gareis et al. Please still proofread the whole manuscript.

Once, more thank you for an interesting paper and good luck with the improvements, that I believe need only to be minor.

References:
Aaltonen K. (2011). Project stakeholder analysis as an environmental interpretation process. International Journal of Project Management. Vol. 29, Iss. 2. pp. 165-183.

Author Response

Reviewer and Comment Number

Reviewer Comment

Author Response

Page and Line Number in Manuscript

n/a

Thank you for offering me the opportunity to review this paper,   which I found highly interesting – this piece of work offered me novel ideas   and insights that are useful also in my own work. The reflections on the role   and linkages between project sustainability management and stakeholder   thinking were valuable, as well as the ideas on utilizing attribution theory   to understand the sensemaking and interpretation processes of project   managers and stakeholders. What I also valued was the fact that you aim to   analyze both the perspectives of PMs and stakeholders and through this   understand the complex interactional processes of these “two” groups (which   are, however, many times characterized by high degrees of multiplicity).

Thank   you. The authors believe that this is an important topic which deserves   greater attention from both academics and practitioners, therefore relevant for the Administrative Sciences.

n/a

1 - 1

First of all, I think you could   be more specific with regard to the concept of secondary stakeholders in your   introduction. The definition of secondary stakeholders is not quite   straightforward in prior literature, and you could somehow explicate what   your understanding in this paper is on this concept. I found the discussion   on the importance on the authenticity of stakeholder “management” highly   relevant in your introduction. I was, however, looking for more references to   recent work on stakeholder engagement, which, at least to my understanding,   better captures the management for stakeholder ideals as well as the value   co-creation perspectives.

Thank you very much for raising this important point. The definition   of secondary stakeholders has been clarified and more updated references   pertaining stakeholder engagement have been added in text to better capture   the management for stakeholder ideals.

We really believe we   addressed this helpful comment of the Reviewer and thus improved the paper   accordingly.

Page 2, Line 36

Page 3, Line 62

Page 3, Line 64

Page 4, Line 69

Page 8, Line 164

Page 15, Line 326

Page 17, Line 356

1 – 2

That is to say, I think that although the message of the   failures of complex projects/large infrastructure projects in their   stakeholder management is to certain extent true, I think that the real life   situation might not be that black and white: we have a number of very   successful projects, that have adopted a value co-creation approach with   regard to secondary stakeholders and have delivered excellent results e.g. in   terms of the innovation outcomes. The culture is changing (slowly though),   and even though I can see the importance of highlighting “the failure focus”   from the perspective of justifying the importance and uniqueness of your   study, I think you could also give more recognition to the work that has   focused on the engagement perspective and the positive dynamics and outcomes   within the area of project stakeholder management. Here, potentially   introducing some relevant procedures (e.g. social impact assessment, which as   a consultative process can bring very valuable outcomes, and encourages in early   dialogues with stakeholders) might be of value.

Maybe you could also provide positive perspective and attempts   in terms of stakeholder engagement in your literature review (e.g. in many   projects SIAs are a highly useful and value producing arena for early   stakeholder engagement and consensus building). At the moment, the picture   that is portrayed on the state-of-the-art/practice of secondary stakeholder   management is relatively negative. However, I do agree with you that prior   research has mostly focused on the negative side and failure cases.

Thanks for this very   constructive and insightful comment. This helped us to provide and recognize   also the positive perspectives highlighted in recent years (E.g. through the   NETLIPSE study). Nonetheless, to recognize some of the successful procedures   implemented in order to achieve a better value co-creation approach with   regard to secondary stakeholders (e.g. Social Impact assessment and Statutory   planning act).

Page 16, Line 346

1 – 3

There is also some work at least on the way project managers   perceive, interpret and enact their stakeholder environment, which has   claimed that also PMs differ in their interpretation and approaches toward   external stakeholders (Aaltonen, 2011). This relates closely to your   attribution approach and may be useful for you.

Many thanks for suggesting such   relevant paper, which has now been included in our paper and strengthened our   argumentations.

Page 4, Line 80

Page 27, Line 580

1 – 4

You are also stating that “no research has discussed…”. In   today’s world, full of researches and studies, we can rarely be certain that   “no research has discussed”. Maybe you could be more modest here and milden   your statement a bit e.g. that “to the best of our knowledge, no research..”.

Thank   you, the sentence has been edited accordingly.

Page 4, Line 87

1 - 5

It would also be really great, if you could clearly explicate   your research question in the Introduction section.

Many thanks. The introduction section culminates with a clear research   question.

Page 5, Line 95

Page 31, Line 648

1 – 6

The literature review is well written and delivers many valuable   ideas and thoughts. Taking the holistic theme of the study and the dual focus   on stakeholder management and sustainability, it is understandable that it is   a bit fragmented. It could also be, that you could focus more clearly on the   theme of stakeholder engagement, cognitive processes and attribution theory   in particular. In this scenario, the review of e.g. stakeholder theory could   play a significantly minor role. However, on the other hand, I enjoyed   reading your “sensemaking” on stakeholder literature, which was fresh and   provided interesting insights.

Thank   you very much for this constructive comment, which helped us to improve the   work and overcome theoretical fragmentation. The focus has been given to the   holistic approach to stakeholder management, the cognitive processes and   attribution theory in particular. Section 2.1 has been deleted and paper   reframed in order to sharpener the focus of the article.

Page 5, Line 106

Page 8, Line 175

Page 10, Line 220

Page 13, Line 274

Page 14, Line 309

Page 17, Line 372

1 – 7

While I learned from your literature review, that there is   complexity within/inside the complex groups of stakeholders (referring to the   multiplicity of stakeholders concept) and that more focus should be placed on   individuals, identities, interests (and maybe also potentially on issues that   drive stakeholders), I was not able to identify this thinking so strongly in   your final model (Figure 3). Your model, at least to me, also seemed to   reinforce the attribution of the groups in to their role-based/function-based   roles (project managers and local communities).  This is maybe still   something that you could re-consider.

Thanks   for this very constructive and insightful comment. Despite the Figure 3 (now   Figure 2) might read as a role/function based “group”, the real contribution   of the paper/model is to understand the cognitive process which takes into   account the needs, expectations and perceptions of individual stakeholders.   The figure shows the dynamics that often operate at the local community   level, where different groups with different needs and expectations have   historically been negatively affected or have affected the unsustainable project   developments. The models shows, how these negative dynamics might be shifted   to positive ones following the normative management-for-stakeholder principles   claiming for a caring organization approach, which aims to inclusion,   fairness and value creation at the local level of MPIC projects. Above all,   speaking of inclusiveness and communication at the local level underpins the   idea of dealing with stakeholders as faceless groups which is the prevalent   approach against stakeholders. Instead, it is founded on the feminist   re-reading of stakeholder theory that, on its own, calls for a new approach   towards stakeholders based on their individual characteristics, demands and   emotions. Therefore, what Figure 3 is delivering is exactly the opposite. It   calls for opening the black box of local community and dealing with the   individuals inside that.

Page 28, Line 596

Page 29, Line 612

1 – 8

I very much liked your ideas on applying the attribution theory   in order to facilitate our understanding of the interpretation processes of   project personnel and local communities/secondary stakeholders. This   perspective reinforces and follows the research stream that has called for   more understanding on the actual cognitive processes of decision-makers and   stakeholders alike. I would, however, still urge you to reconsider the   conceptual framework. I can see the point in this positive/negative   categorization, but what about the different shades of grey, as well as   interactions between these processes during the project lifecycle? I think   that the relationships and interpretations can also be multi-faceted and even   issue specific within projects and both negative and positive collaborative   dynamics (and interpretations) may co-exist at the same time. What kind of   implications would this kind of approach have for your model? Maybe this is   something that you could also consider in the areas for future research.    Your Figure 3 also seems to have some kind of system dynamics thinking   behind it (the re-enforcing cycles). Is this a correct interpretation and   should it be made more visible and explicit?

Thank   you very much for this constructive advice. This has been included into the   recognized limitations of the study and calling for further areas of   research. Nevertheless, it is so true that the communication with stakeholder   individuals is a multi-faceted approach. This is even more complex in MPICs   where the complexities of the project and the decision making among numerous   stakeholders makes the condition more challenging. However, this is not   something to be generalizable, not even from one project to another one.   That, as Philips et.al. (2003) indicated is not and should not be the role of   stakeholder theory to bring a day-t-day manual for dealing with stakeholders.   Such an approach should be in contradiction with the ultimate aim of   stakeholder theory. While stakeholder theory, and any other theory under that   umbrella such as the one providing here, provides some moral principles for   dealing with stakeholders, it leaves the practical decisions to practitioners   so that they deal with the complexities of the individuals’ culture, emotion   and concerns in each case. This, however, would not limit the researchers   from exploring the opportunities to extend stakeholder theory with the   support of other psychological and social theories. These theories would   provide some principals that are less abstract than stakeholder theory but   they have the same limitation of not bringing practical decision processes. Therefore,   future research in the stakeholder management must be done under umbrella of   stakeholder theory, and just like any other theoretical contribution path,   must be supported by empirical data. But considering the limitation of   dealing with complexities in the nature of humans, they are limited to   proposing or confirming principles and not providing suggestions for dealing   with individuals. Stemmed from this, our future research will be analyzing   individuals in a case, and bringing some principles for dealing with   individual humans in other similar contexts.

Page 28, Line 596

Page 31, Line 647

Page 32, Line 672

1 – 9

Finally, there are some typos in terms of the references e.g. it   should be Gareis et al. Please still proofread the whole manuscript.

Thanks   for these concluding, overall comments. We believe that, by responding in   detail to each one of the previous comments, we fixed the issues that are   finally pointed out here. Thank you again to the Reviewer for all of her/his   precious advice throughout the whole manuscript. By addressing each one, we   feel that the work has been really improved. Thanks again for your efforts   and guidance.

Page 16, Line 341

Reviewer 2 Report

Congratulations.

It was a real pleasure to read the paper.

I am always afraid to read a paper (books) which tells about the theory of management (without examples, practical solution proposals). I afraid even more when I can see "sustainable" word.

You - Authors - has broken my threat. Thank you.

The problem is widely and clearly introduced.

Findings and proposals are clearly, logically proved.

The content really reflects what the title suggests.

I'm impressed how the term "sustainability" is explained in relation to the result of the project execution, as well as to the process of the project execution (what was really surprised for me).

Some minor remarks:

The conclusion should have number 7 (not 5)

I doubt if "Project managers have mainly focused on the project as a unit of analysis..." (Line 231). I see their main activity as decision making (based sometimes on analysis). I think that every time project managers can see the project as a process that should end with success. It is their role to manage the whole project. The holistic approach is widening to the sustainability issues and minor stakeholders (to make them well informed).

(Line 292) "Projects are drivers of change..."

I think that you mean MPIC or another type of megaproject. Writing a new accounting software is a project too.

These are my thought to be considered (not must be considered)

Dear Authors. Congratulations once more. Beautiful english language, clear logical content. Impressive.

Author Response

(The authors gave the same response as above.)

Author Response

Page and Line Number in Manuscript

2 - 1

Congratulations.

It was a   real pleasure to read the paper.

I am   always afraid to read a paper (books) which tells about the theory of   management (without examples, practical solution proposals). I afraid even   more when I can see "sustainable" word.

You -   Authors - has broken my threat. Thank you.

The   problem is widely and clearly introduced.

Findings   and proposals are clearly, logically proved.

The   content really reflects what the title suggests.

I'm   impressed how the term "sustainability" is explained in relation to   the result of the project execution, as well as to the process of the project   execution (what was really surprised for me).

Thank you. The authors   believe that this is an important topic which deserves greater attention from   both academics and practitioners, therefore relevant for   the Administrative Sciences.

n/a

2 - 2

The   conclusion should have number 7 (not 5) 

Thank   you. After revising the paper conclusions is now n.6

Page 31, Line 647

2 – 3

I doubt if "Project managers have mainly focused on the   project as a unit of analysis..." (Line 231). I see their main activity   as decision making (based sometimes on analysis). I think that every time   project managers can see the project as a process that should end with   success. It is their role to manage the whole project. The holistic approach   is widening to the sustainability issues and minor stakeholders (to make them   well informed).

Thanks for   raising this point, which has been edited accordingly.

Page 10, Line 221

2 – 4

(Line   292) "Projects are drivers of change..." I think that you mean MPIC   or another type of megaproject. Writing a new accounting software is a   project too.

Many   thanks.

Page 13, Line 276

2 – 5

These are   my thought to be considered (not must be considered)

Dear   Authors. Congratulations once more. Beautiful English language, clear logical   content. Impressive.

Thanks   for these concluding, overall comments. We believe that, by responding in   detail to each one of the previous comments, we fixed the issues that are   finally pointed out here. Thank you again to the Reviewer for all of her/his   precious advice throughout the whole manuscript. By addressing each one, we   feel that the work has been really improved. Thanks again for your efforts   and guidance.

n/a

Reviewer 3 Report

Introduction

The first two sentences do not really link up well. You started with large-scale infrastructure and construction but ‘’jump’’ to public infrastructure and construction project with showing any statement(s) that link the two together. You also moved down to make a statement at line 63 about facility development project-how does this link to the research context or what are the differences between the three concepts? A facility is also more broader and it could mean a lot of things and does not sit well in the context where it is used. Further, to my understanding infrastructure is not exactly the same as construction projects (the former is more broader whilst the latter is more specific; thus construction projects are a form of infrastructure; hence, use ‘’and’’ to put the two together makes it quite confusion. You may consider starting from infrastructure in general, then move to construction infrastructure; then public sector construction (which is also infrastructure). This will help the put the context into its right perspective.

In line 64, you made a strong claim ‘’ The majority of prior project research has focused on the management of those primary stakeholders important to the project's resources’’ without any citation to back claim. Further, what project are you referring to here, I understand you maybe referring to MPIC projects but you don’t have to assumed that the reader should know, especially when that statement is in the different paragraph from where the MPIC is mentioned. ‘’Majority’’ also means that extensive and therefore the citations should be relative extensive to back claims.

Key component that should be espoused extensively is the ‘’negative bond’’ that exist in the implementation of MPIC as it’s the main centre of the research, however, this is scantily presented in the introduction-only two sentences is devoted to at last but two paragraph. I think you should lay more emphasis on this negative bond and/or provide a ‘’definition or description’’ of it before the full review about it is done in the literature.

In the contribution/justification section (last but one paragraph), you referred to two ‘’key’’ stakeholders (community and project organisation/performing organisation) to make your claims but in earlier statements, you referred to more stakeholders (check line 62-64, you referred to other ‘’secondary and external groups …’’). Can you reconcile the two arguments.  

Literature review

Freedman’s also attribute the first idea of stakeholder theory to internal memo of Stanford Research Institute (SRI) in 1963. You may add this.

Further, there has been recent debate about stakeholder engagement rather than management as authors and practitioners argue that ‘’you’’ cannot manage stakeholders and therefore you should engage them.

Overall, this section is really extensive and well-articulated. However, the literature does not link to the ‘’negative bond’’ really strongly. You may consider; At the end of each section or towards the end should have a link to the main issue (negative bond) in the study. Or there could be a sub-section devoted to ‘’stakeholder conflict’’ where opposing stakeholders could create a negative bond that impact on the implementation and performance of MPIC project. On the basis of this, you findings could address how these negative bond will turn into positive. You can also read about stakeholder coalition in project management, where due to stakeholder conflict, weaker stakeholders can form ‘’coalitions’’ to positively or negatively affect projects.  

Methodology

This section is missing.

You should have a section here outlining the procedure that you followed throughout the study; more especially how the literature review was conducted. You also need to provide the philosophical assumptions and approaches/strategies of your study. You may consider the following work: Bryman, A. and Bell, E. 2015. Business research methods. 4th ed. Oxford: Oxford Univ. Press.; Khan, K.S., Kunz, R., Kleijnen, J., Antes, G., 2003. Five steps to conducting a systematic review. Journal of Royal Society of Medicine,  96, 118–121.; Lu, W., Liu, J., 2014. Research into the moderating effects of progress and quality performance in project dispute negotiation. International Journal of Project Management, 32, 654–662.; Ke, Y., Wang, S., Chan, A., Cheung, E., 2009. Research trend of public–private partnership in construction journals. Journal of Construction Engineering Management. 135, 1076–1086.

Analysis/Discussions

This section needs to be re-visited. The presentation of argument in most part of this section sound more of a literature review  than an analysis and discussions of the findings. Probably, you may consider integrating the managerial implications into the discussions and move part of the literature review in the analysis and discussion sections into the Literature Review/Conceptual section.

In essence, the analysis and discussions should look to discuss the already stated findings in the literature review section by teasing out the theoretical and practical (managerial) implications of the findings. Then, this is followed by the development of the framework.

Conclusions

General comments

Author Response

(The authors gave the same response as above.)

Author Response

Page and Line Number in Manuscript

3 - 1

Introduction:

The   first two sentences do not really link up well. You started with large-scale   infrastructure and construction but ‘’jump’’ to public infrastructure and   construction project with showing any statement(s) that link the two together

Thank   you.  Consistency has been applied   throughout the text.

n/a

3 - 2

You also moved down to make a statement at line 63 about   facility development project-how does this link to the research context or   what are the differences between the three concepts? A facility is also more   broader and it could mean a lot of things and does not sit well in the   context where it is used. Further, to my understanding infrastructure is not   exactly the same as construction projects (the former is more broader whilst   the latter is more specific; thus construction projects are a form of   infrastructure; hence, use ‘’and’’ to put the two together makes it quite   confusion. You may consider starting from infrastructure in general, then   move to construction infrastructure; then public sector construction (which   is also infrastructure). This will help the put the context into its right   perspective.

Thanks for this very   constructive and insightful comment. The term ‘facility’ has been deleted in   text in order to ensure logical consistency, avoid confusion to the reader   and put the context into its right perspective. The term Major public   infrastructure and construction project (MPIC) has been used throughout the   text.

This is in accordance with   Flyvbjerg (2014) and Jia et al., (2011) claiming that Infrastructure spending is mainly driven by major projects (i.e.   megaprojects), where many more and larger Public Infrastructure and   Construction projects are being proposed and introduced   (Flyvbjerg, 2014).

Page 2, Line 58

3 - 3

In line 64, you made a strong claim ‘’ The   majority of prior project research has focused on the management of those   primary stakeholders important to the project's resources’’ without any   citation to back claim. Further, what project are you referring to here, I   understand you maybe referring to MPIC projects but you don’t have to assumed   that the reader should know, especially when that statement is in the   different paragraph from where the MPIC is mentioned. ‘’Majority’’ also means   that extensive and therefore the citations should be relative extensive to   back claims.

Thanks for raising this point, which has been addressed accordingly.   Recent citations based on literature reviews in project and stakeholder   management have been added to support our claim.

Moreover, we do refer to any projects here, despite their size,   budget, and/or geographical location. By definition, a projects has limited   resources and, historically, project research has focused on the management of those ‘primary’   stakeholders important to these often constrained resources.

Page 3, Line 54

Page 3, Line 62

3 - 4

Key   component that should be espoused extensively is the ‘’negative bond’’ that   exist in the implementation of MPIC as it’s the main centre of the research,   however, this is scantily presented in the introduction-only two sentences is   devoted to at last but two paragraph. I think you should lay more emphasis on   this negative bond and/or provide a ‘’definition or description’’ of it   before the full review about it is done in the literature.

Thanks   for this very constructive and insightful comment. The “negative dynamics”   often occurring at the local level of MPIC have been better presented by   introducing the importance of perceptions into the stakeholder management   field (this creates a better link to the attribution theory later   introduced). The negative dynamic between local community and project   organization is a common theme throughout the paper, and many examples of   local oppositions have been highlighted by extent studies in the project   management arena. Moreover, for the scope of this paper, the “negative bond”   is explained and in-depth elucidated in the analysis and discussion section   (4.1).

Page 3, Line 54

Page 4, Line 73

Page 4, Line 80

Page 15, Line 328

Page 16, Line 349

Page 17, Line 360

Page 20, Line 440

3 - 5

In the   contribution/justification section (last but one paragraph), you referred to   two ‘’key’’ stakeholders (community and project organisation/performing   organisation) to make your claims but in earlier statements, you referred to   more stakeholders (check line 62-64, you referred to other ‘’secondary and   external groups …’’). Can you reconcile the two arguments.

Thank   you for raising this point. A better linkage and logical argumentation has   been made.

Page 7, Line 163

3 – 6

Literature:

Freedman’s also   attribute the first idea of stakeholder theory to internal memo   of Stanford Research Institute (SRI) in 1963. You may add this.

Thank   you. This section has been removed, according to Reviewer 1.

n/a

3 – 7

Further, there has been recent debate about   stakeholder engagement rather than management as authors and practitioners   argue that ‘’you’’ cannot manage stakeholders and therefore you should engage   them.

Thanks for this comment,   which has been considered by the authors. According to extant literature,   stakeholder management is an essential contributing element to better project   performance. Stakeholder management provides a solid basis for stakeholder   identification, classification and assessment (Cleland, 1986; Donaldson and   Preston, 1995; Eskerod et al., 2015; Olander, 2007; Sutterfield et al.,   2006), which are the first steps required for effective stakeholder   engagement (Reed, 2008).

We do not disagree with the   reviewer on the fact that ‘we’ cannot really manage the stakeholders; however   we believe that an effective engagement is not possible if a structured   managerial processes and activities (identification, classification, and   assessment) are not in place.

n/a

3 - 8

Overall, this section is really extensive and well-articulated.   However, the literature does not link to the ‘’negative bond’’ really   strongly. You may consider; At the end of each section or towards the end   should have a link to the main issue (negative bond) in the study. Or there   could be a sub-section devoted to ‘’stakeholder conflict’’ where opposing   stakeholders could create a negative bond that impact on the implementation   and performance of MPIC project. On the basis of this, you findings could   address how these negative bond will turn into positive. You can also read   about stakeholder coalition in project management, where due to stakeholder   conflict, weaker stakeholders can form ‘’coalitions’’ to positively or   negatively affect projects.

Literature, findings and conclusions have been   reviewed and clearly presented in terms of structure and contents. A better   link to the concept of ‘negative bond’ has been crated throughout the paper.   Statements are linked to the work carried out. A clear definition and   clarification of theoretical and practical outcomes are shown in a logical   flow including the contribution for both academics and practitioners   elucidating how these negative bonds can be turned into positive.

Page 5, Line 106

Page 17, Line 372

Page 20, Line 438

Page 28, Line 596

Page 31, Line 647

3 – 9

Methodology

This section is missing.

You should have a section here outlining the procedure that you   followed throughout the study; more especially how the literature review was   conducted. You also need to provide the philosophical assumptions and   approaches/strategies of your study. You may consider the following work:   Bryman, A. and Bell, E. 2015. Business research methods. 4th ed. Oxford:   Oxford Univ. Press.; Khan, K.S., Kunz, R., Kleijnen, J., Antes, G., 2003.   Five steps to conducting a systematic review. Journal of Royal Society of   Medicine,  96, 118–121.; Lu, W., Liu, J., 2014. Research into the   moderating effects of progress and quality performance in project dispute   negotiation. International Journal of Project Management, 32, 654–662.; Ke,   Y., Wang, S., Chan, A., Cheung, E., 2009. Research trend of public–private   partnership in construction journals. Journal of Construction Engineering   Management. 135, 1076–1086

Many thanks for suggesting such relevant works that although constructive,   are not in line to the objectives of our study. In fact, this paper does not   aim to provide a systematic literature review, which yes requires a   solid methodology section in order to justify the inclusion and exclusion   criteria and the way arguments are constructed and presented.

In fact, conceptual papers   typically do not have methodology section (E.g. Alvesson and Sandberg, 2011;   Crane and Ruebottom, 2012; Cropanzano, 2009; Driscoll and Statik, 2004;   Dunham et al., 2006; Eskerod et al., 2015; Flyvbjerg et al., 2009; Frooman,   1999; Hart and Sharma, 2004; McVea and Freeman, 2005; Rowley, 1997;   Sanderson, 2011; Soderlund, 2004; Sutton and Staw, 1995), instead we have   tried to use theory and concepts to construct our arguments on how we have   arrived at this problem.

“In general, conceptual and   theoretical manuscripts do not have methodology sections. There is no argument being made   that the broad scope of a body of literature has been explored and new   findings are emerging from an analysis” (Callan, 2010, p.302).

Instead, according to Callan   (2010), we have choosing key pieces of literature that support a particular   perspective that we are putting forth for consideration.

Following   top-ranked journals in business and management studies, we strongly believe   that a methodology section is not needed for the typology of work presented   here, which strongly differ from a systematic review.

n/a

3 – 10

Analysis/Discussions

This section needs to be re-visited. The   presentation of argument in most part of this section sound more of a   literature review than an analysis and discussions of the findings. Probably,   you may consider integrating the managerial implications into the discussions   and move part of the literature review in the analysis and discussion   sections into the Literature Review/Conceptual section.

In essence, the analysis and discussions   should look to discuss the already stated findings in the literature review   section by teasing out the theoretical and practical (managerial) implications   of the findings. Then, this is followed by the development of the framework.

Thanks for this very constructive   advice. We made important amendments in the discussion section to have   sharper contribution. Managerial implications have been merged into the   analysis and discussion section in order to strengthening our reflection. We,   however, need to emphasize that the explanations of the shift from negative   bond to the positive bond is the contribution of this article emerged through   the discussion of the literature conceptualized through the lens of   attribution theory and the feminist reading of stakeholder theory. This has   helped to advance current understanding of the normative formulation of the   stakeholder theory and has contributed to develop our final conceptual   framework.

By considering your precious   suggestions, we reviewed again the discussion section of the paper and we   believe now that this section brings enough conceptualizations to rigorously   support its final claims.

Page 20, Line 438

Page 24, Line 508

Page 26, Line 556

Round 2

Reviewer 3 Report

You need to work on the introduction and methodogy 

Author Response

Dear Reviewer,

Thank you very much for your precious comments, which have been addressed or considered accordingly.

INTRODUCTION:

A better linkage between concepts has been made.

Please refer to Page 2, Lne 44; Page 2 Line 55. Page 2, Line 67.

METHODOLOGY: 

Many thanks for suggesting such relevant works that although constructive, are not in line to the objectives of our study.

In fact, this paper does not aim to provide a systematic literature review, which yes requires a solid methodology section in order to justify the inclusion and exclusion criteria and the way arguments are constructed and presented.

Bryman and Bell (20150 is a business research methods book, very familiar to the authors as we use this book for our postgraduate students. Although constructive, it only provides basic differences between systematic and narrative literature review in order to engage with what other have written. The other suggested literature is related to systematic approach, which is not employed in this study.

On the other hand, in line with high ranked ABS Journals (2*, 3*, and 4*), we reinforce the fact that conceptual papers typically do not have methodology section (E.g. Alvesson and Sandberg, 2011; Crane and Ruebottom, 2012; Cropanzano, 2009; Driscoll and Statik, 2004; Dunham et al., 2006; Eskerod et al., 2015; Flyvbjerg et al., 2009; Frooman, 1999; Hart and Sharma, 2004; McVea and Freeman, 2005; Rowley, 1997; Sanderson, 2011; Soderlund, 2004; Sutton and Staw, 1995 - and many more). Following well established literature, we have tried to use theory and concepts to construct our arguments on how we have arrived at this problem.

Meaningful is the work of Callan (2010) here: 

“In general, conceptual and theoretical manuscripts do not have methodology sections. There is no argument being made that the broad scope of a body of literature has been explored and new findings are emerging from an analysis” (Callan, 2010, p.302).

Instead, according to Callan (2010), we have choosing key pieces of literature that support a particular perspective that we are putting forth for consideration.

Following top-ranked journals in business and management studies, we strongly believe that a methodology section is not needed for the typology of work presented here, which strongly differ from a systematic review.

Thank you.